# Targeting the Id1-Kif11 Axis in Triple-Negative Breast Cancer Using Combination Therapy

**DOI:** 10.3390/biom10091295

**Published:** 2020-09-08

**Authors:** Archana P. Thankamony, Reshma Murali, Nitheesh Karthikeyan, Binitha Anu Varghese, Wee S. Teo, Andrea McFarland, Daniel L. Roden, Holly Holliday, Christina Valbirk Konrad, Aurelie Cazet, Eoin Dodson, Jessica Yang, Laura A. Baker, Jason T. George, Herbert Levine, Mohit Kumar Jolly, Alexander Swarbrick, Radhika Nair

**Affiliations:** 1Cancer Research Program, Rajiv Gandhi Centre for Biotechnology, Kerala 695014, India; archanapt@rgcb.res.in (A.P.T.); reshmamurali@rgcb.res.in (R.M.); nitheeshpvk@gmail.com (N.K.); binitha.a.v@gmail.com (B.A.V.); 2Manipal Academy of Higher Education (MAHE), Manipal, Karnataka 576104, India; 3The Kinghorn Cancer Centre and Cancer Research Theme, Garvan Institute of Medical Research, Darlinghurst, NSW 2010, Australia; weeteo2011@gmail.com (W.S.T.); a.mcfarland@garvan.org.au (A.M.); d.roden@garvan.org.au (D.L.R.); h.holliday@garvan.org.au (H.H.); c-v-k@hotmail.com (C.V.K.); a.cazet@garvan.org.au (A.C.); e.dodson@garvan.org.au (E.D.); j.yang@garvan.org.au (J.Y.); laura_baker11@hotmail.com (L.A.B.); a.swarbrick@garvan.org.au (A.S.); 4St Vincent’s Clinical School, Faculty of Medicine, University of New South Wales, Darlinghurst, NSW 2052, Australia; 5Center for Theoretical Biological Physics, Rice University, Houston, TX 77005, USA; jtg6@rice.edu (J.T.G.); herbert.levine@rice.edu (H.L.); 6Medical Science Training Program, Baylor College of Medicine, Houston, TX 77005, USA; 7Departments of Bioengineering and Physics, Northeastern University, Boston, MA 02115, USA; 8Centre for BioSystems Science and Engineering, Indian Institute of Science, Bangalore 560012, India; mkjolly@iisc.ac.in

**Keywords:** cancer stem cells, chemoresistance, self-renewal, combination therapy, *Id1*, *Kif11*

## Abstract

The basic helix-loop-helix (bHLH) transcription factors inhibitor of differentiation 1 (*Id1*) and inhibitor of differentiation 3 *(Id3)* (referred to as *Id*) have an important role in maintaining the cancer stem cell (CSC) phenotype in the triple-negative breast cancer (TNBC) subtype. In this study, we aimed to understand the molecular mechanism underlying *Id* control of CSC phenotype and exploit it for therapeutic purposes. We used two different TNBC tumor models marked by either *Id* depletion or *Id1* expression in order to identify *Id* targets using a combinatorial analysis of RNA sequencing and microarray data. Phenotypically, Id protein depletion leads to cell cycle arrest in the G0/G1 phase, which we demonstrate is reversible. In order to understand the molecular underpinning of Id proteins on the cell cycle phenotype, we carried out a large-scale small interfering RNA (siRNA) screen of 61 putative targets identified by using genomic analysis of two Id TNBC tumor models. Kinesin Family Member 11 (*Kif11*) and Aurora Kinase A (*Aurka*), which are critical cell cycle regulators, were further validated as Id targets. Interestingly, unlike in *Id* depletion conditions, *Kif11* and *Aurka* knockdown leads to a G2/M arrest, suggesting a novel *Id* cell cycle mechanism, which we will explore in further studies. Therapeutic targeting of *Kif11* to block the *Id1–Kif11* axis was carried out using small molecular inhibitor ispinesib. We finally leveraged our findings to target the *Id/Kif11* pathway using the small molecule inhibitor ispinesib in the Id+ CSC results combined with chemotherapy for better response in TNBC subtypes. This work opens up exciting new possibilities of targeting *Id* targets such as *Kif11* in the TNBC subtype, which is currently refractory to chemotherapy. Targeting the *Id1–Kif11* molecular pathway in the Id1+ CSCs in combination with chemotherapy and small molecular inhibitor results in more effective debulking of TNBC.

## 1. Introduction

Breast cancer is a heterogeneous disease with different molecular subtypes displaying distinct outcomes [1,2]. The triple-negative breast cancer (TNBC) subtype does not express molecular markers such as estrogen receptor (ER) and human epidermal growth factor receptor 2 (HER2) that are the basis of targeted therapies in other molecular subtypes of breast cancer [3,4]. Consequently patients presenting with TNBC are left with few therapeutic choices, resulting in lower five-year survival rates when compared to the other subtypes [3]. Hence, there is an urgent need to understand the molecular basis of TNBC in order to identify new drug targets.

The critical role of a subpopulation of cells termed cancer stem cells (CSCs) in self-renewal, chemoresistance, and metastasis has assumed clinical importance in breast cancer [5,6]. The inhibitor of differentiation (Id) proteins are negative regulators of the basic helix-loop-helix (bHLH) transcription factors [7]. The Id proteins are important for maintaining the CSC population and therefore tumor progression in TNBC. We have previously shown that *Id1/3* (collectively known as *Id*) are critical for the CSC-associated phenotypes in the TNBC molecular subtype [1]. Genetic screen analysis of *Id* knockdown (*Id* KD) and *Id1* expression models led to the identification of *Kif11* and *Aurka* as putative Id targets in this study.

The detailed mechanism by which *Id* controls the cell cycle is not clear, although Id is known to impact the pathway via decreased expression of cyclins *D1* and *E*, reduced phosphorylation of *Rb* as well as reduced cyclin E*-Cdk2* activity [8]. In this work, we show how *Id* acts as a central focal point to coordinate the cell cycle genes *Kif11* and *Aurka* and demonstrate that *Id* KD leads to cell cycle arrest in the G0/G1 phase of the cell cycle. Interestingly, we found that the depletion of *Kif11* and *Aurka* independently did not phenocopy the G0/G1 arrest we observed in *Id* KD cells. We demonstrated that *Id* KD puts the brakes on the cell cycle, resulting in a state of arrest at the G0/G1 phase via impacting cell cycle molecules. Moreover, we demonstrated that *Id* is a critical driver of self-renewal, acting via *Kif11* and *Aurka*. We found that the Id-expressing tumor cells were resistant to chemotherapy, which forms the first line of treatment in TNBC. Interestingly, treatment with ispinesib, a small molecule inhibitor against *Kif11*, resulted in the reduced expression of Id in these cells. We finally exploited this finding to treat tumor cells with the chemotherapeutic drug paclitaxel, combined with ispinesib to ablate the Id-expressing chemoresistant tumor cells along with bulk tumor cells, leading to more effective therapeutic targeting in the TNBC subtype.

## 2. Methods

### 2.1. Cell Culture 

4T1 cells were sourced from American Type Cell Culture Collection (ATCC) and cultured as per specifications. The 4T1 Id1GFP cells were generated by lentiviral infection of the 4T1 cells with the Id1 GFP reporter, as reported previously [1]. 

### 2.2. Immunofluorescence

Cells were fixed in 4% paraformaldehyde (TCL119-100ML, Himedia, Mumbai, India), permeabilized with 0.2% TritonX-100 (T9284, Sigma-Aldrich, St. Louis, Missouri, USA), and blocked with 1% bovine serum albumin (BSA) (A7906, Sigma-Aldrich) in phosphate buffered saline (PBS) (Gibco, Grand Island, NY, USA)for 1 h at room temperature. The cells were then incubated with primary antibody at 4 °C overnight, followed by secondary antibody for 1 h at room temperature. The nuclei were stained with 4′,6-diamidino-2-fenylindool (DAPI; Sigma-Aldrich). Images were taken by confocal microscopy (Leica Microsystems, Wetzlar, Germany).

### 2.3. Cell Cycle Analysis 

Cells were harvested and incubated with Hoechst 33342 (H3570, Invitrogen, CA, USA; 4 ug/mL) at 37 °C for 30 min. The cell cycle distribution was determined with a flow cytometer (BD FACS Aria III, BD Biosciences, San Jose, CA, USA). The data were analyzed using the BD FACS analyzer software.

### 2.4. Quantitative Real-Time PCR (qRT-PCR)

Total RNA was isolated using TRIzol Reagent (15596026, Invitrogen, Carlsbad, CA, USA)), and complementary DNA (cDNA) was synthesized using High-Capacity cDNA Reverse Transcription Kits (4368814, Applied Biosystems, Foster City, CA, USA). Real-time PCR was performed on the QuantStudio 7 Flex Real-Time PCR System (Applied Biosystems) using Power SYBR^®^ Green PCR Master Mix (Applied Biosystems). All target gene expression levels were normalized to β-actin. The relative fold change was determined by 2^−ΔΔCT^ method, as described previously [9]. The sequences of primers used to detect target mouse genes in qRT-PCR are listed in Appendix A.

### 2.5. Flow Cytometry

Adherent cells were trypsinized and washed with PBS and blocked with Fc Receptor (FcR) blocking reagent (130-092-575, Miltenyi Biotec, Bergisch Gladbach, Germany) for 1 h. The cells were then incubated with fluorescein isothiocyanate (FITC)-conjugated anti-mouse cluster of differentiation 29 (CD29) (130-102-975, Miltenyi Biotec) and R-phycoerythrin (PE)-conjugated anti-human cluster of differentiation 24 (CD24) (130-103-371, Miltenyi Biotec) antibodies (1:50 dilution) for 20 min at 4 °C in the dark. Cells were washed twice with PBS and resuspended in 500 µL of fluorescence activated cell sorting (FACS) buffer prior to analysis on a FACS flow cytometer (BD FACS Aria III, BD Biosciences).

### 2.6. Microarray and Bioinformatics Analysis

Total RNA from the samples were isolated using a Qiagen RNeasy min ikit (Qiagen, Doncaster, VIC, Australia). cDNA synthesis, probe labelling, hybridization, scanning, and data processing were all conducted by the Ramaciotti Centre for Gene Function Analysis (The University of New South Wales). Gene expression profiling was performed using the AffymetrixGeneChip Gene 1.0 ST Array, a whole-transcript array that covers 28,000 coding transcripts and 7000 non-coding long intergenic non-coding transcripts. Data analysis was performed using the Genepattern software package from the Broad Institute. Three different modules, Hierarchical Clustering Viewer, Comparative Marker Selection Viewer, and Heatmap Viewer were used to visualize the data. In addition to identifying candidate molecules and pathways of interest, Gene Set Enrichment Analysis (GSEA) (http://www.broadinstitute.org/gsea) was performed using the GSEA pre-ranked module. Briefly, GSEA compares differentially regulated genes in an expression profiling dataset with curated and experimentally determined sets of genes in the MSigDB database to determine if certain sets of genes are statistically over-represented in the expression profiling data.

### 2.7. siRNA Screen to Assess Proliferation

Reverse transfection of 4T1 cells in 384-well plates was performed with 400 cells and 0.08 µL Dharmafect1 per well using a Caliper Zephyr and Biotek EL406 liquid handling robots. Media was changed at 24 h post-transfection. CellTiter-Glo^®^ assay (Promega, Madison, WI, USA) was performed using a BMG Clariostar plate reader (luminescence assay). 4T1 cells were reverse-transfected with a 40nM siGENOMESMART pool siRNA against each of the 57 candidate genes. Cell viability was quantified at 72 h post-transfection using the CellTiter-Glo^®^ assay (Appendix A). IncuCyte ZOOM live cell imaging every 2 h was also performed, which allowed us to quantify cell growth (confluence) over time throughout the experiment.

Final data presented were generated from three biological replicates, each consisting of two technical replicates. Viability measurements were normalized to the treatment-matched scrambled control after subtracting the blank empty wells.

### 2.8. Western Blotting

Cells were lysed with radio immunoprecipitation assay (RIPA) buffer (R0278-50ML, Sigma-Aldrich) containing 1X complete protease inhibitor cocktail (P8340-5ML, Sigma-Aldrich). Protein concentrations were determined using Pierce™ BCA Protein Assay Kit kit (23225, Thermofisher scientific, Waltham, Massachusetts, USA). Proteins were separated by sodium dodecyl sulfate polyacrylamide gel electrophoresis (SDS-PAGE) and transferred onto 45 μm polyvinylidene difluoride (PVDF) membrane (1620177, Biorad, Hercules, California, USA). Membranes were blocked using 5% non-fat dry milk in Tris-buffered saline with 0.1% Tween 20 (P1379, Sigma) and probed with the respective primary and secondary antibodies. The signal was detected based on enhanced chemiluminescence (ECL) using Clarity Western ECL Substrate (1705061, Bio-Rad, Berkeley, CA, USA. β-actin was used as a protein loading control. Antibodies used in this study are listed in Appendix A.

### 2.9. MTT Assay

4T1 cells were seeded at a density of 500 cells per well in a 96-well plate. When the cells became 80% confluent, we added freshly prepared 3-(4,5-Dimethylthiazol-2-yl)-2,5-Diphenyltetrazolium Bromide (MTT) reagent (M6494, Thermofisher Scientific, Hyderabad, India) (5 ug/mL) to the culture media. The plates were incubated for 2 h in the dark at 37 °C. The media-containing reagent was removed from each well and 100 µL of DMSO was added to each well. Absorbance reading was taken using a TECAN microplate reader at 570 nm.

### 2.10. IC_50_ Values for Chemotherapeutic Drugs

4T1 cells were harvested and seeded 1000 cells per well in a 96-well plate. When the cells became 20% confluent, culture media was removed and replenished with media containing the chemotherapeutic drugs paclitaxel (10 nM) and doxorubicin (30 nM); small molecular inhibitors ispinesib (5 nM), alisertib (100 nM), and MTT reading was taken 48 h post-drug treatment.

### 2.11. Tumorsphere Assay

4T1 cells were put into the tumorsphere assay, as described previously [1]. Paclitaxel and ispinesib were added at a concentration of 10nM and 5nM, respectively, to assay for the effect of these drugs on self-renewal.

### 2.12. The Kaplan–Meier Plotter

The association of *ID1* (OMIM accession number: 600349) and *KIF11* (OMIM accession number: 148760) gene expression with the relapse-free survival (RFS) of breast cancer patients was analyzed using the Kaplan–Meier Plotter (KM Plotter, Semmelweis University, Budapest, Hungary) database (http://kmplot.com/analysis/) [10]. The gene expression and survival data are derived from Gene Expression Omnibus (GEO). KM Plotter is a manually curated database handled by the PostgreSQL server that integrates gene expression with clinical data. The patient cohorts were split on the basis of median gene expression by auto-select best cut-off. The *ID1* and *KIF11* genes were entered into the KM Plotter database and the RFS was determined for the TNBC subtype (*n* = 198 patients). To perform multiple gene analysis, the mean expression of *ID1* and *KIF11* were used. The hazard ratio (HR) with 95% confidence and *p*-values were obtained from the KM Plotter. *p*-values <0.05 were considered as statistically significant.

### 2.13. Statistical Analysis

Statistical analyses were performed using GraphPad Prism 6 (GraphPad Software, Inc., San Diego, CA, USA). All in vitro experiments were performed in three biological replicates each with two or more technical replicates. Data represented are means ± standard deviation. Statistical tests used are unpaired Student’s *t*-test and two-way ANOVA. *p*-values < 0.05 were considered statistically significant with * *p* < 0.05, ** *p* < 0.01, *** *p* < 0.001, **** *p* < 0.0001.

## 3. Results

### 3.1. Id Depletion Leads to a G0/G1 Cell Cycle Arrest that Is Reversible

It has been previously demonstrated that *Id* KD significantly affects pathways associated with cell cycle progression [8,11]. We first sought to validate this observation in our model system using the pSLIK construct [12,13]. We used the metastatic 4T1 cell line as it is representative of the TNBC subtype [14], and Id proteins have been shown to play an important role in tumorigenesis in the TNBC subtype. We used an inducible known down system, as reported earlier [1], and observed a significant decrease in the proliferative capacity of cells upon doxycycline (Dox)-induced *Id* KD in comparison to control conditions.

As proliferation is inextricably linked to the cell cycle, we next characterized the effect of *Id* KD on cell cycle progression. We found that *Id* depletion resulted in G0/G1 arrest, as seen in a significant increase in the G0/G1 fraction when compared to the controls (Figure 1a–c). To further elucidate the molecular mechanism through which *Id* controls the cell cycle, we analyzed the effect of *Id* KD on the expression of key cell cycle genes that are vital at different phases of the cell cycle. The downregulation of *Id* significantly decreased the expression of Cyclin A2 (*Ccna2*), Cyclin B1 *(Ccnb1*), Cyclin B2 (*Ccnb2*)*,* Cyclin-dependent kinase1 (*Cdk1*), and *c-Myc*, as shown in Figure 1d. Interestingly, we found an inverse correlation with *Rb* and *p21*, which are the negative regulators of these cell cycle genes (Figure 1e).

### 3.2. Identification of Putative Id-Regulated Genes

To characterize the network of genes regulated by Id proteins, we performed functional annotation analysis on gene array and RNA sequencing data from two different TNBC models of tumor cells marked by either *Id* depletion or Id1 expression (Figure 2a). The gene array analysis of the *Id1* depletion system has been described previously [1]. The Id1 expression model analyzed genes whose expression was associated with *Id1*, whereas the *Id* depletion model attempted to identify downstream targets of Id proteins.

To study the phenotypes associated with depletion of *Id* as well as to assess its downstream targets, we compared the gene expression profile of three independent replicates of control and *Id* KD cells by microarray analysis to generate a list of differentially expressed genes between *Id-*depleted and control cells (Appendix A). Aiming to discover high confidence genes involved in the *Id* gene regulatory network, we compared lists of differentially expressed genes in the *Id*-depleted TNBC model and their controls using MetaCore™ software. The data were uploaded in MetaCore and filtered using an adjusted *p*-value threshold of 0.05, resulting in 4301 network objects that were differentially expressed between the *Id* KD and the control cells (Figure 2a). To characterize the network of genes regulated by *Id*, we performed enrichment analysis on the candidate genes, which was visualized by process networks and pathway maps. In addition, we used the Id1C3Tag model system to prospectively isolate Id1+ cells, as described previously [1]. The gene expression profiles of the Id1+ and Id1- cells from three independent Id1C3Tag tumors were compared by RNA sequencing exclusively in this study. This resulted in a list of differentially expressed genes between the Id+ and Id- mouse TNBC cells (Appendix A). Similarly, the Id1C3Tag RNA sequencing data revealed 126 network objects differentially expressed between Id1+ and Id1- cells (Figure 2a). Interestingly, when we looked at the pathway analysis generated from the differentially expressed gene lists of both models, cell cycle pathways were among the top hits (Appendix A).

By comparing these two datasets, we generated lists of MetaCore network objects common to both experiments as well as those unique to each of the two datasets (Figure 2b). When comparing these lists of network objects from the two TNBC models, we identified 34 high confidence MetaCore network objects as common to both the datasets of differentially expressed genes. Finally, the genes were mapped to network objects in MetaCore, resulting in a list of 26 genes that were significantly regulated in both models of TNBC (Table 1).

We first analyzed the pathways controlled by the 26 putative *Id* targets. Interestingly, the main pathways regulated by *Id* involved the cell cycle, specifically the metaphase checkpoint, spindle assembly, and chromosome segregation (Figure 2b,c). Disruption of checkpoint control and aberrant regulation of the cell cycle are observed in tumorigenesis [7,15], resulting in uncontrolled cell proliferation. A key function of *Id* is the stimulation of cell cycle progression and proliferation by controlling the activity of cell cycle regulators. Studies have already reported that overexpression of *Id* has been associated with upregulated cell cycle progression in tumorigenesis [7,8,11]. Pathways involving cytoskeleton remodeling, integrin-mediated cell adhesion, migration, and chemotaxis, which are all key steps in epithelial to mesenchymal transition (EMT) and metastasis, were also enriched. Analysis of each individual experiment, along with the genes common to both datasets, showed a similar result, with *Id* depletion mainly affecting the cell cycle pathway, DNA damage, checkpoints, and cytoskeleton remodeling. *Id1* expression model alone showed enrichment for pathways involving hypoxia-induced epithelial–mesenchymal transition, *WNT* pathway in development, cytoskeleton remodeling, and cell cycle (Figure 2c).

We also looked at the Epithelial-Mesenchymal Transition (EMT) program, which is an important driver of the CSC state, and interestingly found a change in both the E-cadherin and vimentin protein levels (Appendix A). However, when we analyzed the EMT scores of these samples using an inferential scoring metric [16], it did not show any significant change, indicating that the EMT status of the cells was not altered upon *Id* KD (Appendix A).

### 3.3. Identification of Kif11 and Aurka as Potential Id Targets

Among the 26 differentially expressed genes common to the two TNBC models, we prioritized 16 genes for validation. These were chosen on the basis of a significant *p*-value (<0.05) and at least 1.5-fold up- or downregulation compared to the controls. Most of these genes had opposite regulation in the two TNBC models, which was consistent with the fact that one model was marked by *Id* depletion whereas the other was an *Id1* expression model. In addition, eight potential cancer stem cell markers and genes previously implicated in Id biology were chosen on the basis of cell surface localization, significant enrichment in Id+ cells, and availability of antibodies. Altogether 61 candidate genes were identified for further validation as putative *Id* candidate target genes (Appendix A).

We next went on to assess the role of the candidate targets of Id proteins on the proliferative phenotype using a targeted siRNA screen. Interestingly, the target genes that showed the greatest effect on the viability and thus the proliferative phenotype of the *Id* KD cells with more than 50% were *Kif11, Casc5, Ccnd1*, and *Aurka* (Figure 3a, Appendix A). The hits at the other end of the scale included Robo1, which has been investigated in our work separately [1].

To confirm putative Id targets such as *Kif11*, *Aurka*, *Ccnd1*, and *Casc5*, we analyzed relative mRNA levels. We observed significant reduction in the mRNA levels of *Kif11*, *Aurka*, Cyclin D1 *(Ccnd1*), and Cancer susceptibility candidate 5 (*Casc5*) in *Id* KD compared to the controls (Figure 3b). We also detected a decrease in the expression of *Kif11*, *Aurka*, *Ccnd1*, and *Casc5* at the protein level by Western blot (Figure 3c,d). *Kif11* and *Aurka* were also downregulated at the transcript level in tumorspheres generated in the *Id* KD cells when compared to control (Figure 3e, Appendix A).

To confirm the effect of putative *Id* targets on proliferation, we used an independent pMission siRNA system in 4T1 parental cells. Loss of *Kif11* and *Aurka* lead to significant decrease in the proliferative capacity of the 4T1 cells (Figure 3f, Appendix A). We continued our studies with *Kif11* and *Aurka* as we did not observe any significant loss of proliferative phenotype with *Casc5* and *Ccnd1* knockdown (Appendix A), although we did observe a decrease at the mRNA level (Appendix A).

### 3.4. Kif11 or Aurka Depletion Did Not Phenocopy Loss of Id

We next characterized the effect of *Kif11* KD and *Aurka* KD on the cell cycle using the pMission system. We found that *Kif11* and *Aurka* depletion lead to cell cycle arrest in the G2/M phase, as evidenced by cell cycle analysis (Figure 4a–c). Intriguingly, this observation was fundamentally different from Id KD case, where it undergoes G0/G1 arrest (Figure 1a, Appendix A).

To determine the molecular effect of *Kif11* and *Aurka* on cell cycle, we analyzed relative mRNA levels of key cell cycle genes. *Kif11* KD significantly reduced the expression of Aurka, even though the expression of *Id1* was not altered (Figure 4d). *Kif11* depletion had a positive effect on the mRNA levels of *Id3, Casc5, Rb*, and *p21* and reduced the expression of *Cdk1*, but had no significant effect on *Ccnd1, c-Myc*, and *p53*. *Aurka* KD did not show any effect on *Id1, Id3, Kif11,* or *Ccnd1*, while there was an increased expression of *Casc5* and *p21* (Figure 4e). We next compared our microscopic observations on the phenotype of the Id KD system with that of *Kif11* KD and *Aurka* KD. We noticed monoastral bodies with improperly assembled mitotic spindle, indicating that the majority of the *Kif11* KD and *Aurka* KD cells were arrested in the M phase of the cell cycle (Figure 4f, Appendix A). The formation of monoastral bodies is indicative of duplicated chromatin (4N) not being able to separate due to perturbations in the spindle formation and centrosome, thus indicating G2/M arrest and matching the cell cycle analysis [17] (Figure 4a). Quantification of percentage mitotic cells (Figure 4g) and cells exhibiting monoastral body phenotype (Figure 4h) in *Kif11* KD and *Aurka* KD clearly demonstrated that Id depletion resulted in a phenotype that is distinct from *Kif11* and *Aurka*.

### 3.5. Therapeutic Targeting of CSCs Through Id–Kif11 Axis

There is currently no effective targeted therapy for TNBC, and chemotherapy is usually the first line of treatment, with a relapse rate of 25% [18]. Our previous work has demonstrated that *Id* is critical for CSC-associated phenotypes in TNBC such as proliferation, self-renewal, migration, and metastasis [1]. We have now identified that these phenotypes are influenced by the *Id-Kif11/Aurka* axis. We hypothesized that targeting *Kif11* or *Aurka* to block the *Id1-Kif11/Aurka* axis may cause the *Id*-expressing CSC to be more vulnerable to chemotherapy and more effectively debulk the entire tumor mass.

To test this hypothesis, we first determined the IC_50_ values for two commonly used chemotherapy drugs in breast cancer treatment, paclitaxel and doxorubicin, in 4T1 cells (Appendix A). Interestingly, we found that there was a significant enrichment for Id1+ tumor cells after treatment with paclitaxel and doxorubicin, suggesting that the Id1+ CSCs are chemoresistant (Figure 5a,b). We next determined the IC_50_ values for the small molecule inhibitors of Kif11 and Aurka, ispinesib, and alisertib (Appendix A). Cells treated with ispinesib showed a significant reduction in the percentage of Id1+ cells but no significant change was observed in those treated with alisertib as compared to the control (Figure 5c). We decided to continue with paclitaxel and the *Kif11* inhibitor, ispinesib, on the basis of these results.

We next asked whether ispinesib can increase the sensitivity of Id1+ cells to paclitaxel. We found that the cells treated with a combination of paclitaxel and ispinesib showed a significant decrease in cell viability as well as Id1 and Kif11 expression when compared to paclitaxel or ispinesib alone (Figure 5d, Appendix A). Interestingly, we found that the protein level expression of Kif11 was also significantly reduced when treated with the combination (Appendix A).

We next checked the effect of combination therapy on the self-renewal phenotype. A significant reduction in the self-renewal capacity was observed in cells treated with the combination of paclitaxel and ispinesib when compared to either of the drugs alone (Figure 5e,f). To conclude, we have demonstrated that treatment with conventional chemotherapeutic drugs such as paclitaxel enriched Id+ cancer stem cells, which could result in tumor relapses in patients in the clinic. Inhibition of the *Id* target Kif11 (with ispinesib treatment) in combination with chemotherapy also resulted in a loss of Id+ CSC subpopulations of tumor cells, ultimately leading to more effective debulking of the entire tumor (Figure 5g).

The correlation between *ID1* and *KIF11* gene expression and relapse free-survival of breast cancer patients was analyzed by the KM Plotter database. TNBC patients with a higher mean gene expression of *ID1* and *KIF11* together had poor prognosis (HR- 1.91, *p*-value < 0.0082) when compared to patients with low expression of both genes (Appendix A). This work opens up the possibility of using the *KIF11* inhibitor to improve prognosis in patients having with expression of both *ID1* and *KIF11* using a combination of chemotherapy and small molecule inhibitors such as ispinesib.

## 4. Discussion

The current body of work sheds light on the role that Id proteins (specifically Id1 and Id3) play in affecting key CSC phenotypes such as proliferation and self-renewal through multiple mechanisms. We observed a striking G0/G1 cell cycle arrest when cancer cells were depleted of Id proteins. By regulating the expression of critical cell cycle genes, *Id* pauses or checks the cells in the G1 state in a manner that they can re-enter the normal cell cycle once the stress is removed [1]. This supports the theory of *Id* as a master regulator that, on sensing unfavorable conditions, can “brake” the cells in the G1 phase through multiple means (molecular regulation of cell cycle genes, DNA division inhibition, protein complex perturbation at the centrosome, and spindle fibers). This strategy would allow cells to survive in a state of stasis until conditions favorable to growth of the tumor cell set in.

The idea that CSCs are more plastic and can exist in more than one state may be supported by looking at the Epithelial-Mesenchymal Transition (EMT) program [18]. From the point of view of the EMT scores, we observed that the *Id* KD is not pushing the cells clearly towards a more E or a more M state, as the levels of both canonical markers decreased. Moreover, our bioinformatic model used a ratio of Ecadherin (*CDH1)*/ Vimentin (*VIM)* as a predictor to calculate the scores; thus, the relatively proportional change that we saw at RNA/protein levels for both CDH1 and VIM was consistent. The data adds to the evidence that EMT and Mesenchymal-Epithelial Transition (MET) are not binary [19] for different stages of EMT and their varying degrees of causal contribution to metastasis.

Using two independent models of *Id1* gene expression and gene depletion, we were intrigued to identify the critical cell cycle genes, *Kif11* and *Aurka*, as *Id* putative gene targets. Previous work in nasopharyngeal cells has linked *Id1* and *Aurka* mechanistically in the induction of tetraploidy. Id1 was found to affect Aurka degradation, which normally occurs during exit from mitosis by the APC/C Cdh1-mediated proteolysis pathway. Id1 stabilized Aurka by active competition with Cdh1, thus preventing Aurka degradation [20]. Interestingly, while individual knockdown of *Kif11* and *Aurka* also led to a proliferative arrest, it did not phenocopy the G0/G1 cell cycle arrest with the *Id* knockdown or the formation of monoastral bodies. This suggested that the impediment of the cell cycle by Id protein is through different mechanisms and not the canonical mitotic pathways involving the microtubules by *Kif11*/ *Aurka*, which forms a part of our future investigation.

There is no targeted therapy for TNBC, and chemotherapy is the first line of treatment. Thus, we checked the effect of the commonly used chemotherapy drugs paclitaxel and doxorubicin, which are used in the clinic for treatment of TNBC. Studies by our group and others have already reported that Id1 marks a chemoresistant breast cancer cell [7] in cancers such as hepatocarcinomas [21], and the Kif11 pathway has been targeted in the treatment of docetaxel-resistant TNBC cells [22]. However, the most compelling reason to target the *Id1*/*Kif11* pathway came from work [23] that identified the drug BRD9876 as a kinesin-5 inhibitor in multiple myeloma, which led to significant downregulation of *ID1*. On the basis of our work, we used Id as a marker for the chemoresistant CSC population in TNBC. We tested our hypothesis that we can achieve a better response by combining traditional chemotherapy along with ablation of the Id-expressing chemoresistant cells using small molecule inhibitors against the Id target Kif11. We achieved a significant decrease in proliferative and self-renewal capacity when the cells were treated with paclitaxel and ispinesib by successfully targeting subpopulations of cells, including the Id+ CSCs within a tumor.

A key limitation of our study is that we focused on two genes on the basis of the phenotype affected. Future work involves dissecting the role of other putative Id targets along with combinatorial therapy using a wider range of commonly used chemotherapeutic drugs on the basis of the Id targets validated.

Thus, a combination of targeted drugs with chemotherapy would be an effective strategy for the complete treatment of TNBC and would give women currently living with this disease a better long-term prognosis.

## 5. Conclusions

CSCs are drivers of tumor progression and metastasis in breast cancer. Using two model systems with a combined transcriptomic analysis and large scare siRNA screen, we demonstrated the important role of the Id proteins in the cell cycle process, which is corrupted in a cancer context by *Id* targets such as *Kif11* and *Aurka*. This has importantly led to the identification of the Kif11 small molecule inhibitor ispinesib to eradicate the Id1+ cells that are resistant to paclitaxel. Demonstrating the effectiveness of combination chemotherapy with targeted drug therapy against the Id1+ CSC opens up new avenues to explore. Future work involves dissecting the role of other putative Id targets along with combinatorial testing with chemotherapeutic drugs to more effectively debulk the entire tumor cell population.

## 6. Data Availability

The datasets generated for this study can be found in the GEO Database GSE129790, GSE129858, and GSE129859.

## Figures and Tables

**Figure 1 biomolecules-10-01295-f001:**
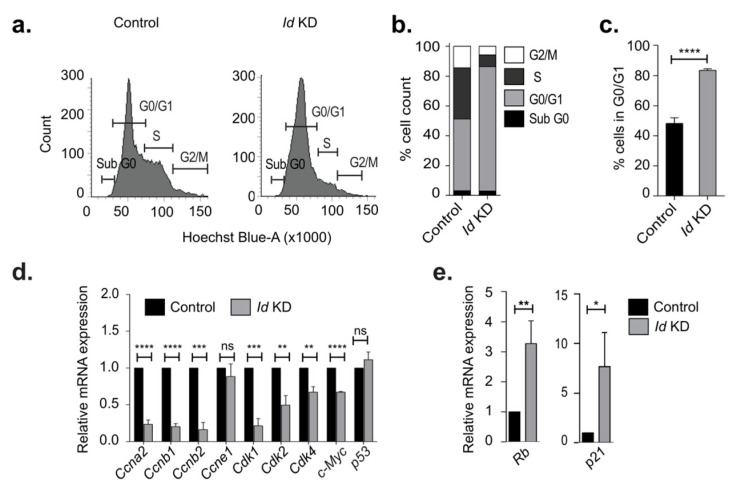
Effect of inhibitor of differentiation (*Id*) knockdown on cell cycle genes. (**a**) Flow cytometric analysis of cell cycle on control and *Id* knockdown (KD) cells after labelling with Hoechst 33342 stain. (**b**) Comparing the percentage cell count in each phase of the cell cycle, showing a significant increase in the G0/G1 phase in *Id* KD conditions. (**c**) Percentage cell count in G0/G1 phase of control and *Id* KD cells. (**d**) The relative mRNA expression level of cell cycle genes Cyclin A2, B1, B2, E1 (*Ccna2*, *Ccnb1*, *Ccnb2, Ccne1*), Cyclin dependent kinase 1, 2, 4 (*Cdk1*, *Cdk2*, *Cdk4*), *c-Myc*, and *p53* in control and *Id* KD cells using qRT-PCR. (**e**) Relative mRNA expression of Rb and p21 in Id KD cells with respect to control cells were quantified using qRT-PCR. Data were normalized to *β-actin* and analyzed by the 2^−ΔΔCt^ method. All experiments were performed in three biological replicates and data were expressed as mean ± standard deviation. Unpaired Student’s *t*-test and two-way ANOVA were used. * *p* < 0.05, ** *p* < 0.01, *** *p* < 0.001, **** *p* < 0.0001, ns is non significant.

**Figure 2 biomolecules-10-01295-f002:**
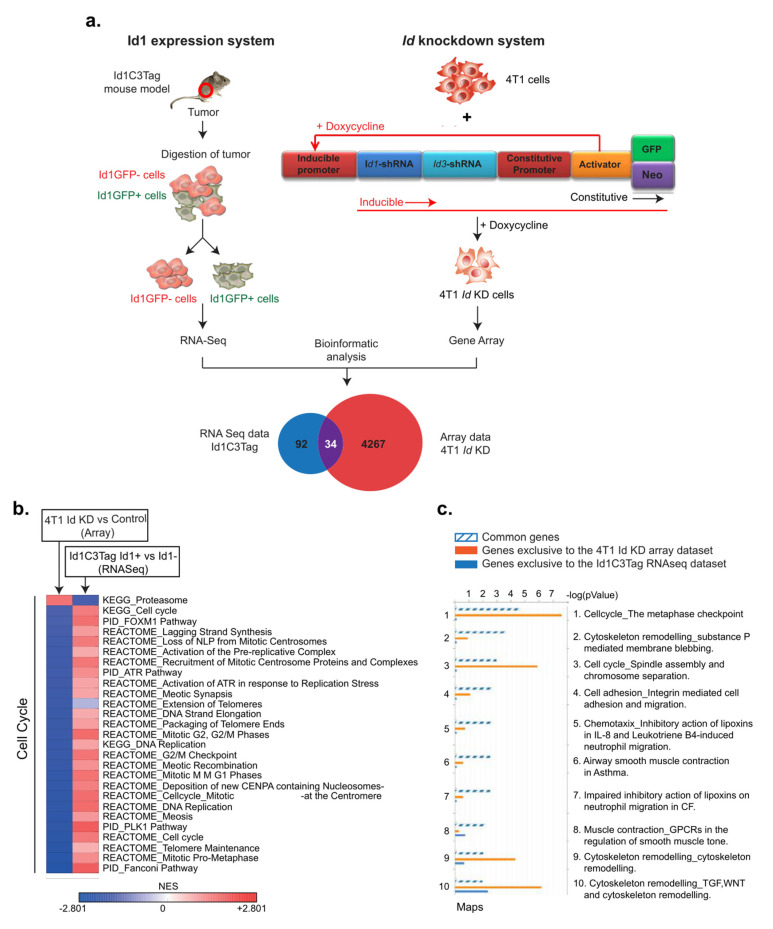
Identification of putative Id-regulated genes. (**a**) To characterize the network of genes regulated by Id proteins, we performed functional annotation analysis on gene array and RNA sequencing data from two different triple-negative breast cancer (TNBC) models marked by either Id depletion or Id1 expression. Aiming to discover high confidence genes involved in the Id gene regulatory network, we compared lists of differentially expressed genes between the Id-expressing or -depleted TNBC models and their controls using MetaCore software. By comparing these two datasets, we generated lists of MetaCore network objects common to both experiments, as well as those unique to each of the two datasets. The data were uploaded in MetaCore and filtered using an adjusted *p*-value threshold of 0.05, resulting in 4301 network objects that were differentially expressed between the Id KD and the 4T1 control cells. Similarly, the Id1C3Tag RNA sequencing data revealed 126 network objects specific to mouse differentially expressed between Id+ and Id- cells. Finally, the genes were mapped to network objects in MetaCore, resulting in a list of 34 genes that were significantly differentially regulated in both models of TNBC. (**b**) To characterize the network of genes regulated by Id, we performed enrichment analysis on the 34 candidate genes, which was visualized by process networks and pathway maps. The enriched pathways included cell cycle, cytoskeleton remodeling, integrin-mediated cell adhesion/migration, and chemotaxis, which are all key steps in epithelial to mesenchymal transition (EMT) and metastasis. (**c**) The cell cycle pathway had the highest score in the genes common to both datasets analyzed, which correlated with the G0/G1 block observed in Id KD cells.

**Figure 3 biomolecules-10-01295-f003:**
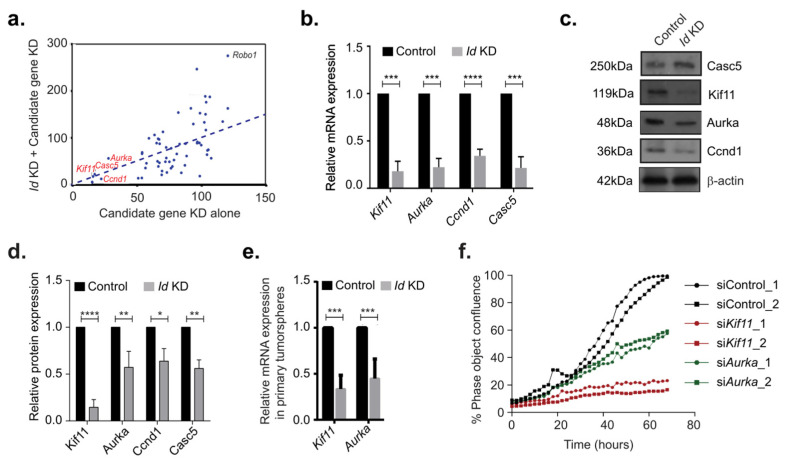
Effect of Id target gene knockdown on the proliferative phenotype. (**a**) The effect of the candidate genes on proliferation of 4T1 cells were assessed by reverse transfection with a 40nM siGENOMESMART pool siRNA against each of the 61 genes. The target genes that showed the greatest effect on the viability and thus the proliferative phenotype of the 4T1 cells with more than 50% were *Kif11*, *Aurka*, *Ccnd1*, and *Casc5*. (**b**) The relative mRNA expression level of *Kif11*, *Aurka*, *Ccnd1*, and *Casc5* in Id KD cells with respect to control cells were quantified with qRT-PCR. Data were normalized to β-actin and analyzed by the 2^−ΔΔCt^ method. (**c**) The expression of the Id target genes *Kif11*, *Aurka*, *Ccnd1*, and *Casc5* at the protein level was decreased in Id KD cells when compared to the controls. (**d**) Quantification of relative protein expression of Kif11, Aurka, Ccnd1, and Casc5 in control and *Id* KD normalized with β-actin. (**e**) Relative mRNA expression level of *Kif11* and *Aurka* in primary tumourspheres. (**f**) Percentage confluency in *Kif11* KD and *Aurka* KD conditions, showing significant decrease in cell proliferation compared to the controls using two independent pMission siRNAs per gene. All experiments were performed in three biological replicates and data were expressed as mean ± standard deviation. Unpaired Student’s *t*-test and two-way ANOVA were used. * *p* < 0.05, ** *p* < 0.01, *** *p* <0.001, **** *p* <0.0001.

**Figure 4 biomolecules-10-01295-f004:**
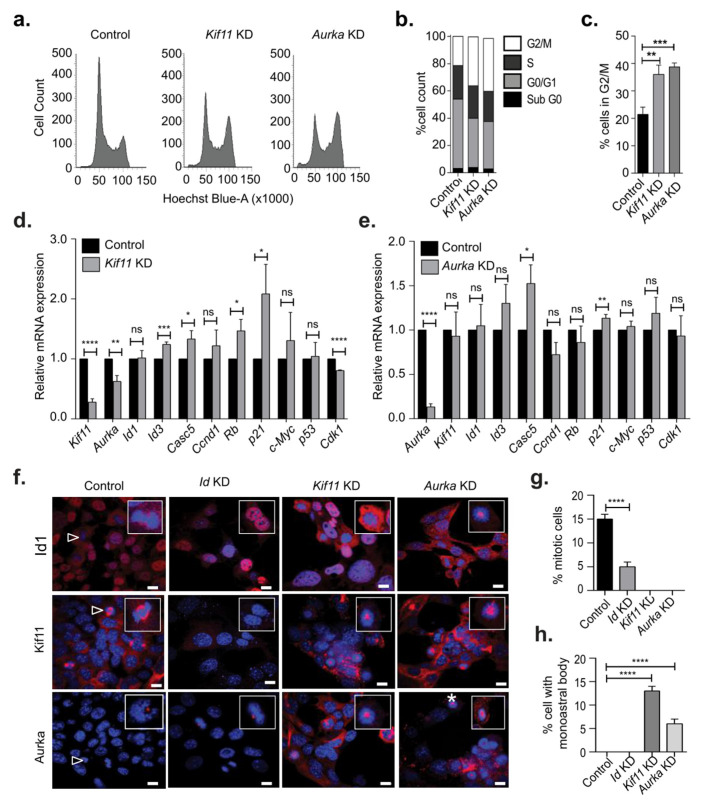
*Kif11* and *Aurka* depletion did not phenocopy loss of Id1. (**a**) Flow cytometric analysis of cell cycle in *Kif11* KD and *Aurka* KD with scrambled control in 4T1 cells. (**b**) Comparing the percentage cell count in each phase of the cell cycle after knocking down *Kif11* and *Aurka*. (**c**) There were significantly higher cells in G2/M phase *Kif11* KD and *Aurka* KD when compared to control. (**d**,**e**) The relative mRNA expression level of important cell cycle gene were analyzed in *Kif11* KD and *Aurka* KD with respect to scrambled control using qRT-PCR. Data were normalized to β-actin and analyzed by the 2^−ΔΔCt^ method. (**f**) Immunofluorescence staining for Id1, *Kif11*, and *Aurka* on 4T1 control, Id KD, *Kif11* KD, and *Aurka* KD cells. Representative images were taken using a Nicon A1R+ confocal system at 100x magnification with a scale bar corresponding to 100um. 4′,6-Diamidino-2-fenylindool (DAPI) shows the nuclear staining; asterisk (*) shows the monoastral bodies formation in siRNA KD system; Δ shows normal cell division; and inset shows the 100× zoomed images of the same images. (**g**) Percentage mitotic cells in control, *Id* KD, *Kif11* KD, and *Aurka* KD cell lines. (**h**) Cells exhibiting monoastral body formation in control, *Id* KD, *Kif11* KD, and *Aurka* KD conditions. All experiments were performed in three biological replicates and data were expressed as mean ± standard deviation. Unpaired Student’s *t*-test and two- way ANOVA were used. * *p* < 0.05, ** *p* < 0.01, *** *p* < 0.001, **** *p* < 0.0001, ns is non significant.

**Figure 5 biomolecules-10-01295-f005:**
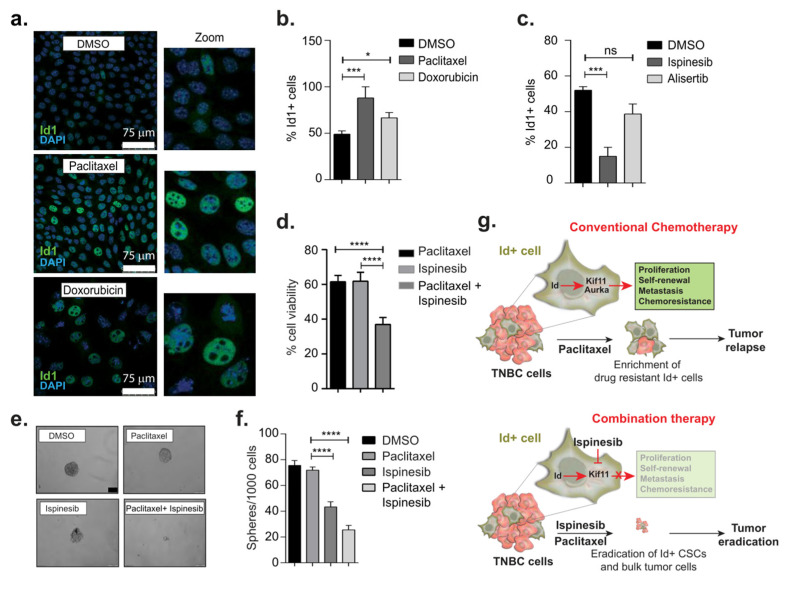
Therapeutic targeting of cancer stem cells (CSCs) through the *Id1-Kif11/Aurka* axis. (**a**) Immunofluorescence images of cells to determine Id1 expression after treatment with paclitaxel and doxorubicin (20×). (**b**) Percentage cells expressing Id1 in control, paclitaxel-, and doxorubicin-treated cells as well as ispinesib and alisertib (**c**) were determined by immunofluorescence. (**d**) Cell viability was determined after treating the cells with paclitaxel, ispinesib, and the combination therapy of paclitaxel + ispinesib. Representative phase contrast images of tumor spheres (**e**) and quantification of number of tumor spheres per 1000 cells (**f**) in control, paclitaxel, ispinesib, and paclitaxel + ispinesib. (**g**) Schematic showing the targeting of *Id–Kif11* axis with conventional and combination therapy leading to more effective eradication of the entire tumor. All experiments were performed in three biological replicates and data were expressed as mean ± standard deviation. Unpaired Student’s *t*-test and two- way ANOVA were used. * *p* < 0.05, *** *p* < 0.001, **** *p* < 0.0001, ns is non significant.

**Table 1 biomolecules-10-01295-t001:** List of differentially expressed genes common to Id1 expression and *Id*-depleted model system. The genes highlighted in green were used in further analysis as putative *Id* targets.

		*Id*_KD_4T1_metacore		*Id1*C3Tag_RNASeq_metacore	
#	Input IDs	Signal	*p*-Value	Signal	*p*-Value
1	***Adamtsl3***	−0.3927	0.020463	1.3402333	0.03359618
2	***Casc5***	−1.0215	0.001402	0.9961643	0.002269021
3	***Aspm***	−1.3871	0.000268	0.9586754	0.000409133
4	***Aurka***	−1.1447	0.0005806	0.9813335	0.01574964
5	***Casz1***	0.2437	0.0479447	0.9948092	0.002230733
6	***Cenpf***	−0.9544	0.001877	0.7718833	0.02742762
7	***Ctla2a***	0.4903	0.0023071	1.3066035	0.04998167
8	***Cxcl15***	2.0181	0.0004083	−6.916629	0.01322961
9	***Angptl7***	−0.631	0.0023779	−1.489744	0.002084568
10	***Cldn6***	−0.5278	0.0131683	−7.315199	0.004137659
11	***Gpr133***	0.284	0.0448178	−4.065336	0.000459154
12	***Hmga1;Hmga1-rs1***	−0.4779	0.0332492	−1.080467	0.001750527
13	***Il6***	1.2691	0.0003497	1.2769696	0.004137659
14	***Kif4***	−1.0266	0.0007331	0.7764488	0.01698252
15	***Kif11***	−1.3438	0.0002452	0.7487605	0.02222188
16	***Lphn1***	0.4649	0.0027409	0.8679558	0.04907909
17	***Ltbp2;Ltbp3***	0.2667	0.0244267	0.9104315	0.002134988
18	***Mylk***	0.5396	0.0154085	1.1309435	0.00340821
19	***Lnp;Nusap1***	−1.011	0.0001961	1.0915483	0.008424267
20	***Pdgfc***	1.7513	0.00001065	0.7934828	0.03592082
21	***Angptl2;Angptl4***	3.5985	5.221 × 10^−7^	−1.456264	1.46715 × 10^−5^
22	***Prc1***	−0.9945	0.0009206	0.7781182	0.02082709
23	***Stc2***	−1.2088	0.0005806	−2.135721	8.46387 × 10^−5^
24	***Mylk***	0.5396	0.0154085	1.1309435	0.00340821
25	***Ube2c***	−0.6684	0.0060464	1.2384745	0.004860396
26	***Upp1***	−0.5451	0.0022003	−1.767675	0.000253826

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
