# Peer review of "Targeting the Id1-Kif11 Axis in Triple-Negative Breast Cancer Using Combination Therapy"

_biomolecules, 2020, doi:10.3390/biom10091295_

Round 1
Reviewer 1 Report
The submitted manuscript by Archana et al. investigated the effects of genetic modulation of the transcription factor Id1 and their role for CSC maintenance and drug resistance of triple negative breast cancer. Several other studies before investigated the role of ID1 for CSCs and demonstrated that 1. Id proteins promote a CSC phenotype in triple negative breast cancer via Robo1-dependent c-Myc activation (Teo et al. 2019, biorvx); 2.) there exist an addiction of breast CSCs to the IGF2-ID1-IGF2 circuit f(Tominaga et a. 2018, oncogene); and 3. that the TGFβ-Id1 Signaling Opposes Twist1 and Promotes Metastatic Colonization Via a Mesenchymal-to-Epithelial Transition (Stankic et al. 2014, Cell Rep).
The present study is well performed, the manuscript is written in a easy to follow style and the interpretation of the results are based on optained results. However, the study was only performed in one murine cell line. The potential of clinical translation is low, because no human cell line or patient material was used to validate these finding. In addition, Id1 is a transcription factor known to regulate cellular fate. I was wondering about the differentiation status of the ID1 KD cells. It would be also crucial to demonstrate direct binding of ID1 to the KLF1 or AurK promotor based on chromatin-immunoprecipitation. Is the ID1 consensus sequence found within the KLF1 or Aurk promotor?
I was also wondering, are there differences in CSC population (phenotype, function, markers) in TNBC compared to other breast cancer subtypes? And is the ID1-dependency only seen in TNBC CSCs?
Minor comments:
- Some appreviation within the abstract, e.g. bHLH were not introduced
- Please upload the gene expression data into public database, e.g. GEO.
- Fig. 1a: add gating within histogram.
- Is ID1 KD regulating ID1 expression themself? How are other IDs affects. Are there compensatory mechanisms between different ID isoforms? I would prefer to always specify within the text which ID isoform was investigated.
- Within the ID1C3Tag model is there still endogeneouse gene expressed or is this depleted? How much is KD and ID1-neg. population overlapping in gene expression? Display differences of genetic and functional approach.
- Fig. 3f: please show the growth curves for siKIF1 and siAurk longer. Do these cells regrow, are they quiescient or do the cells die?
- Fig 4: how is the Kinesin Family Member 11 or Aurora kinase affecting gene expression of Id3? This are no transcription factors and not involved in gene expression regulation. Is the found altered gene expression only correlative and do you have another regulatory upstream mechanisms investigated?
- Fig. 4a: duplicated chromatin (N4) is not seen with Hoechst staining. Why? Please validate with another method or check hypothesis.
- Fig. 4f: only week downregulation of protein upon KD within IF images. Aurka is even higher expressed in KD cells. Is the KD only seen on gene level? How are the functional differences explained?
- Fig.4g: how was mitotic index calculated? Use BrdU, Ki67 or p-H3 staining for validation.
- Paclitaxel and ispinesib have similar molecular targets, the spindle. How do you explain the observed additive effects?
- What are the ID1, KLF and Aurk KD effects on CSC population (marker expression or functional sphere formation)?
Reviewer 2 Report
Please include the OMIM accession number for the important genes discussed, at first mention.
Symbols for genes must be italicized
Please deposit the microarray data in Gene Expression Omnibus (GEO) for Microarray data storage and include the accession number in the manuscript.
Please include cytoscape network images to show the interaction among identified gene/proteins.
Round 2
Reviewer 1 Report
The authors included some of the suggested changes from the reviewer and improved the manuscript. Therefore, I recommend the recent version to be published in Biomolecules.